# Alterations of lower- and higher-order aberrations after unilateral horizontal rectus muscle surgery in children with intermittent exotropia: A retrospective cross-sectional study

Dong Cheol Lee, Se Youp Lee, Jong Hwa Jun[ID]*

Department of Ophthalmology, School of Medicine, Keimyung University, Daegu, Korea

* junjonghwa@gmail.com

## Abstract

**Data Availability Statement:** All relevant data are within the manuscript and its Supporting Information files.

### Background

This retrospective, cross-sectional study investigated changes in corneal lower- and higher-order aberrations that cause visual disturbance after lateral rectus recession and medial rectus resection in children.

### Methods

Eighty-five eyes of 85 patients (44 boys; 8.64±2.88 years) who underwent lateral rectus recession and medial rectus resection to correct intermittent exotropia were assessed. The Galilei G4 Dual Scheimpflug Analyzer was used for wavefront analysis. Risk factors (age, sex, amount of surgery, preoperative axial length, preoperative intraocular pressure) were determined. Outcome measures included simulated and ray-tracing mode keratometry with secondary defocus, oblique, and vertical astigmatism (for lower-order aberrations) and the root mean square, 3rd-order vertical and horizontal coma, oblique and horizontal trefoil, 4th-order spherical aberration, oblique and vertical secondary astigmatism, and oblique and vertical quadrafoil (2nd–8th sums) (for higher-order aberrations).

### Results

Myopic with-the-rule changes in low-order aberrations and increases in simulated and ray-tracing mode keratometry during the 3 months following lateral rectus recession and medial rectus resection were attributed to muscle healing and stability changes. High-order aberrations altered in the week following surgery almost returned to normal within 3 months. Axial length, the amount of surgery, age, and sex affected astigmatism due to differences in patients' scleral states.

**Funding:** This study was supported by the Bisa Research Grant of Keimyung University in 2020 (no URL; no grant number, recipient: DCL). The funder had no role in study design, data collection and analysis, decision to publish, or preparation of the manuscript.

## Conclusions

Clinicians should consider changes in high-order aberrations of young individuals who underwent lateral rectus recession and medial rectus resection and may not be able to verbalize changes in vision.

## Introduction

Intermittent exotropia (IXT) is one of the most common types of strabismus in children and adults and involves one eye intermittently turning out when the individual is tired, with a high prevalence rate (about 68% of all strabismus) in Asian and South African populations [1]. Surgery is a well-established treatment for IXT and aims to reposition the ocular alignment by weakening the lateral recti and strengthening the medial recti to alternate the orientation of their actions [2].

Various reports have evaluated postoperative refractive changes after strabismus surgery. These results included no change [3] and myopic or hyperopic shifts of spherical equivalent [4, 5]. In addition, horizontal muscle surgery was associated with alterations of astigmatism during early postoperative periods [3, 5, 6]. Nevertheless, refractive changes associated with strabismus surgery are temporary [7] and insignificant [4].

In the early postoperative period, visual disturbance are common after lateral rectus recession and medial rectus resection (R&R) in patients with IXT but disappear during a long-term follow-up period. To date, the cause of these postoperative visual symptoms was considered to be an alteration in retinal correspondence by modified alignment or temporary esotropia caused by overcorrection [8]. Nowadays, since the role of aberration changes in visual symptoms is emphasized, it is necessary to objectively confirm subjective visual symptoms, such as diplopia or blurred vision, based on wavefront technology and the corresponding aberration changes.

In this study, we investigated the causes of visual disturbances after R&R surgery for IXT, using the Galilei G4 Dual Scheimpflug Analyzer to investigate lower-order aberrations (LOA) with correction for posterior corneal curvature, as well as higher-order aberrations (HOA)s, and investigated various risk factors affecting postoperative corneal curvature changes in children.

## Methods

### Patient demographics

The study design adhered to the tenets of the Declaration of Helsinki for biomedical research in human subjects and was approved by the Institutional Review Board (No. 2017-01-003) of Keimyung University Dongsan Medical Center. The IRB committee waived the requirement for informed consent due to the retrospective nature of the study, and all data were fully anonymized before accessing them.

We retrospectively reviewed the medical records of all patients (mean age, 8.64±2.88 years) who had undergone R&R surgery for IXT repair, from June 2015 to February 2016, at the Department of Ophthalmology, Dongsan Medical Center, Daegu, Korea. Among these patients, most of them were assessed for refractive error with topography and retinal examination with axial length (AL) before and after R&R surgery to detect the degree of the problem. In this study, we only enrolled patients who underwent Scheimpflug photography with

topography at each visit to evaluate astigmatism and accurately analyze the refractive status. Patients with a history of preceded vertical rectus or oblique muscle surgery, concomitant transposition of horizontal muscles during recession or resection surgery, history of ocular surgery, significant corneal opacity, contact lens wearer, overt allergic or infectious keratoconjunctivitis, or ocular surface diseases within a year that affected the cornea and sclera significantly were excluded.

## Measurements

Preoperative medical records included age, sex, AL, mean angle of exodeviation at distance and near, amount of surgery, best-corrected visual acuity, cycloplegic refraction, and slit-lamp examinations. The angle of exodeviation was measured using the alternate prism cover test for distant (6 m) and near (33 cm) objects using accommodative targets and the patients' best optical correction. Analysis of high- and low-order analysis was performed using the Galilei G4 Dual Scheimpflug Analyzer (Ziemer, Port, Switzerland). The corneal topographic parameters related to simulated or total corneal power were determined by the Galilei G4 Dual Scheimpflug Analyzer. In addition, secondary (second) defocus, oblique, and vertical astigmatism of LOA and third vertical and horizontal coma, oblique and horizontal trefoil, fourth spherical aberration (SA), oblique and vertical secondary astigmatism, and oblique and vertical quadrafoil of HOA were calculated. The total root mean square (RMS) of HOA and summation of each order aberration were also analyzed from the second to the eighth. Each parameter related to the total corneal wavefront analysis was collected under a 6.0 mm pupil diameter condition. The total corneal power, which was measured by the ray-tracing method, was included for considering the posterior corneal astigmatism to analyze the net corneal astigmatism. These values were compared with conventional simulated corneal power and keratometric values. All surgeries were conducted by a single surgeon (LSY) who performed a limbal incision at the medial rectus muscle resection and an incision in the fornix at the lateral rectus muscle recession. Furthermore, patients were divided into two groups based on the extent of the surgery. Patients in the first group underwent unilateral 5.0-mm lateral rectus resection (LR res.), followed by 4.0-mm medial rectus resection (MR res.) and 6.0-mm LR res. with 5.0-mm MR res., whereas patients in the second group underwent 7.0-mm LR res. with 5.5-mm MR res. with a greater extent of the sugery (S1 Table).

## Statistical analyses

R language version 3.3.3 (R Foundation for Statistical Computing, Vienna, Austria) and T&F program ver. 3.0 (YooJin BioSoft, Korea) were used. Data are expressed as mean ± standard deviation. Paired sample t-tests were performed to evaluate the differences in the outcomes between before surgery and at 1 week post-surgery, and before surgery and 3 months post-surgery. When variables were not normally distributed, the Wilcoxon signed-rank test was performed. Outcomes measured before and after surgery, risk factors, such as age, sex, intraocular pressure (IOP), operating volume, AL before surgery, and two time-points were used as paired dependent variables and independent variables, respectively, in the multiple linear mixed-effect model. Time and all risk factors were used as fixed effect covariates with random intercepts and random slopes.

## Results

Eighty-five eyes of 85 patients were finally enrolled; 43 patients (50.6%) were male, and the mean age was 8.64 ± 2.88 years (range: 3–16 years). The average AL of patients was 23.29 ± 1.31 mm. Preoperative mean exodeviation angles were 34 ± 7 prism diopter (PD) at

near and 26 ± 6 PD at far distance. Best-corrected visual acuity was logMAR 0.08 ± 0.66 /0.08 ± 0.72 decimal position. The patients were divided into two groups by age (age < 8 years [32 patients] and ≥ 8 years [53 patients]), AL (< 23.125 mm [42 patients] and ≥ 23.125 mm [42 patients]), and the amount of surgery (5.0 mm: 4.0 mm + 6.0 mm: 5.0 mm versus above). The 5.0 mm: 4.0 mm + 6.0 mm: 5.0 mm group included 51 patients (60%).

## Changes in measurements postoperatively

All preoperative baseline parameters and the changes over time for IOP, refraction, simulated or total corneal power, LOAs, and HOAs are summarized in the S2 Table. Preoperative IOP was 15.60 ± 2.24 mmHg and did not change from baseline to 1-week or 3-months postoperatively. The spherical and cylinder decreased at 1-week (both $p < 0.01$), but slightly increased by 3-months postoperatively, as compared to the 1-week values (both $p < 0.01$) (Fig 1A and 1B).

In the simulated corneal power, the mean, steep K, and astigmatism were slightly increased by 1-week (mean difference: 0.15 ± 0.05, 0.51 ± 0.07, and 0.68 [0.72]; $p < 0.01$, $p < 0.01$, and $p < 0.01$, respectively), although they had returned to baseline by 3-months postoperatively (mean difference: 0.07 [0.35], 0.18 ± 0.05, and 0.25 [0.51]; $p > 0.05$, $p < 0.01$, and $p < 0.01$, respectively). However, flat keratometry (K) decreased at 1-week (-0.31 ± 0.13; $p < 0.05$) and returned to nearly baseline levels by 3-months (-0.19 ± 0.13; $p > 0.05$) (Fig 2C–2F).

In the total corneal power, the mean K and steep K values increased continuously from 1-week ($p < 0.05$, $p < 0.01$, respectively) to 3-months postoperatively ($p < 0.01$). The flat K was decreased by 1-week ($p < 0.01$), although it increased by 3-months postoperatively ($p < 0.01$), as compared to baseline. Although astigmatism was increased at 1-week ($p < 0.01$) after surgery, it slightly decreased, nearly reaching baseline values after 3-months ($p < 0.01$) (Fig 2G–2J).

In particular, when defocus was converted to dioptric (D) value, it decreased to a minus value 1-week postoperatively and increased again at 3-months (preoperative: 0.14 ± 0.32,

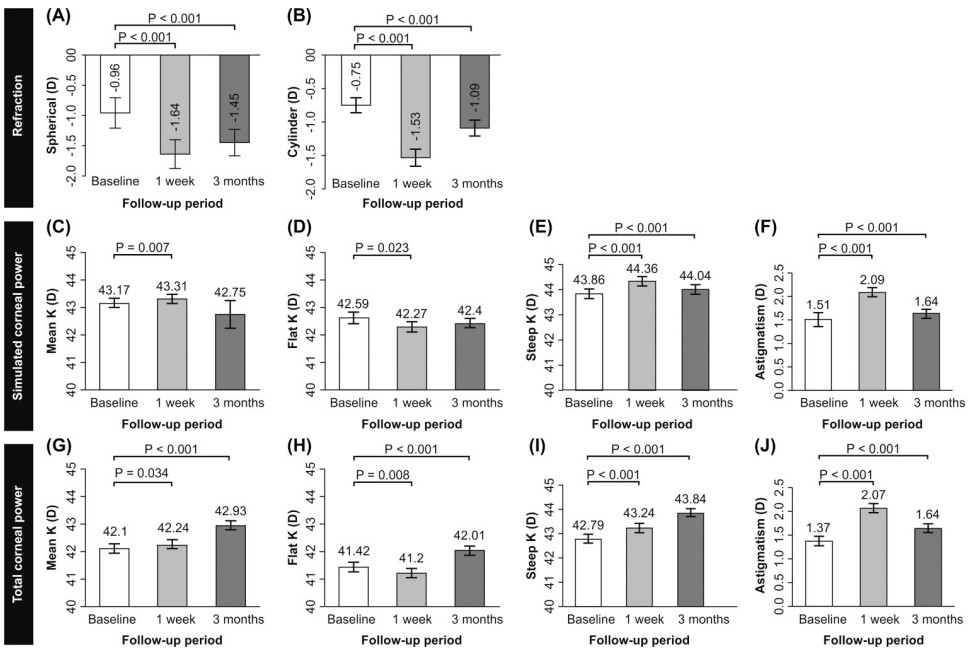

**Fig 1. Change in refraction, simulated and total corneal power according to follow-up time.** Data expressed as mean ± standard error. D: diopter; K: keratometry.

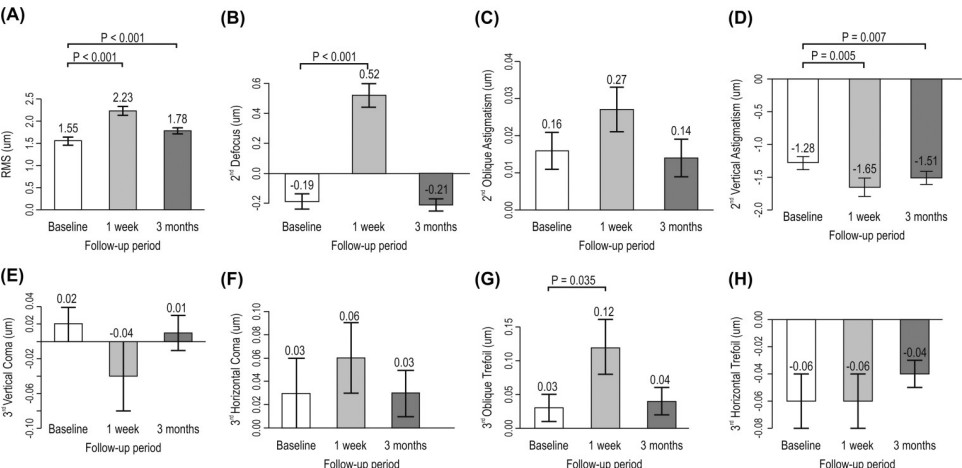

**Fig 2. Change in RMS (μm), second- and third-order aberration (μm) according to follow-up time.** Data expressed as mean ± standard error. RMS: root mean square.

1-week postoperatively: -0.39 ± 0.52, and 3-months postoperatively: 0.26 ± 0.54 diopters, respectively, $p < 0.01$ and $p < 0.05$). However, vertical astigmatism (**μ**m) decreased by 1-week ($p < 0.01$) and increased slightly toward baseline levels after 3-months ($p < 0.01$). Oblique astigmatism did not change from 1-week to 3-months after surgery (Fig 2B–2D).

Overall, the total root square mean (RMS) significantly increased at 1-week ($p < 0.01$), and then recovered toward baseline level after 3-months ($p < 0.01$) (Fig 2A). Secondary order aberrations and defocus were increased by 1-week ($p < 0.01$), and then decreased to nearly baseline levels after 3-months ($p > 0.05$).

In the third-order aberrations, only oblique trefoil increased by 1-week ($p < 0.05$) after operation and returned to baseline level after 3-months ($p > 0.05$) (Fig 3E–3H). In fourth-order aberrations, SA, and vertical quadrafoil were significantly decreased 1-week postoperatively ($p < 0.01$, $p < 0.05$, respectively), although these returned to baseline levels after 3-months ($p > 0.05$). Vertical secondary astigmatism increased 1-week ($p < 0.01$) after the

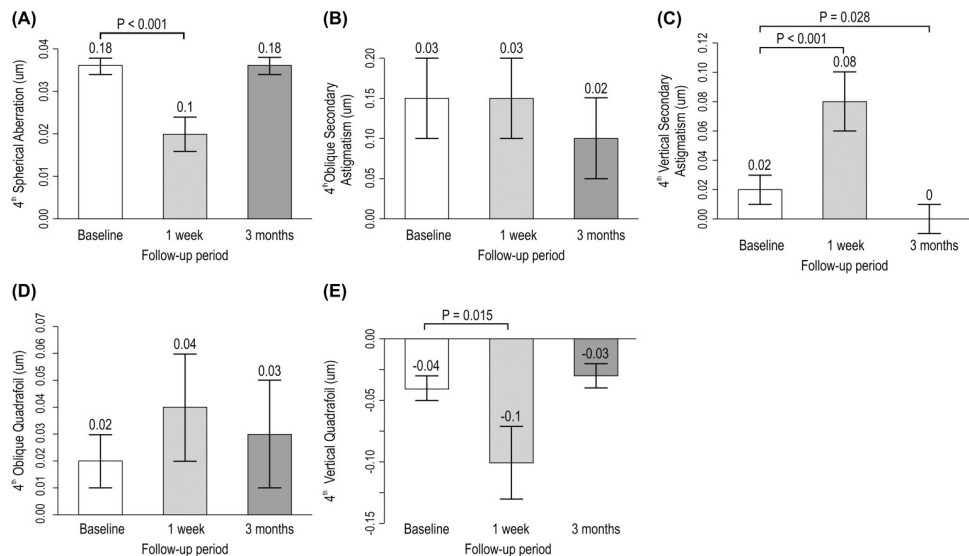

**Fig 3. Change in fourth-order aberration (μm) according to follow-up time.** Data expressed as mean ± standard error.

operation, and then decreased beyond the baseline level after 3-months ($p < 0.05$). The remaining oblique secondary and oblique quadrafoils did not change significantly from 1-week to 3-months postoperatively (Fig 3). In addition, the sum variable of the second to eighth-order aberrations increased by 1-week postoperatively, although the recovery levels remained above baseline, except for second- and fourth- order aberrations (S2 Table).

## Multivariable analysis

In the multiple linear mixed-effects model comparing the preoperative and 1-week postoperative changes, after incorporating factors such as age, sex, AL, IOP, and amount of surgery, we found differences. One week postoperatively, when the AL was $\geq 23.125$ mm, the spherical value decreased significantly as compared to when AL was $< 23.125$ mm ($p < 0.001$). The cylinder value significantly decreased at 1-week postoperatively in those who were aged $\geq 8$ years as compared to those $< 8$ years of age ($p = 0.007$); this value increased in females as compared to males ($p = 0.014$). For total corneal astigmatism, flat, steep, and mean K values were all decreased significantly at 1-week postoperatively when AL was $\geq 23.125$ mm as compared to when AL was $< 23.125$ mm ($p = 0.009$, $p < 0.001$, and $p = 0.003$, respectively). In total corneal astigmatism, patients aged $\geq 8$ years demonstrated significantly increased astigmatism at 1-week after surgery, as compared to those $< 8$ years of age ($p = 0.007$); females showed decreased astigmatism as compared to males ($p = 0.004$). At 1-week after surgery, in patients with AL $\geq 23.125$ mm, the spherical value decreased significantly as compared to those with AL $< 23.125$ mm ($p = 0.032$). In HOAs, the total RMS (μm) in those aged $\geq 8$ years increased significantly at 1-week after surgery, as compared to those aged $< 8$ years ($p = 0.001$); females showed a decreased total RMS compared to males ($p = 0.002$) (Table 1).

**Table 1. Result of multiple linear mixed-effect model analysis of variables.**

| Variables | Delta | Age[a] | Sex[b] | AL[c] | IOP[d] | Amount of surgery[e] |
|---|---|---|---|---|---|---|
| Spherical (D) | 1W –Pre | -0.358 | -0.434 | -2.082 ** | -0.023 | 0.748 |
| | 3M –Pre | -0.342 | -0.371 | -2.013** | -0.139 | 0.830* |
| Cylinder (D) | 1W –Pre | -0.616** | 0.495* | 0.176 | 0.121 | -0.082 |
| | 3M –Pre | -0.421 | 0.550** | 0.070 | 0.130 | -0.125 |
| Flat K[f] (D) | 1W –Pre | 0.028 | 0.438 | -0.902** | 0.251 | -0.214 |
| | 3M –Pre | 0.031 | 0.424 | -0.919** | 0.239 | -0.148 |
| Steep K[f] (D) | 1W –Pre | 0.596 | -0.097 | -1.308** | 0.321 | -0.035 |
| | 3M –Pre | 0.582 | 0.078 | -1.176** | 0.293 | -0.072 |
| Mean K[f] (D) | 1W –Pre | 0.322 | 0.206 | -1.065** | 0.243 | -0.102 |
| | 3M –Pre | 0.313 | 0.248 | -1.040** | 0.246 | -0.093 |
| Astigmatism[f] (D) | 1W –Pre | 0.532** | -0.514** | -0.408* | 0.065 | 0.269 |
| | 3M –Pre | 0.512* | -0.446* | -0.327 | -0.006 | 0.184 |
| RMS total (μm) | 1W –Pre | 0.572** | -0.485** | -0.262 | -0.190 | 0.092 |
| | 3M –Pre | 0.457** | -0.363 | -0.263 | -0.133 | 0.096 |

** $p < 0.01$

* $p < 0.05$, AL: axial length, IOP: intraocular pressure, D: diopter; W: week; M: month; Pre: preoperative; K: keratometry; RMS: root mean square

[a] age $\geq 8$ years

[b] female

[c] AL $\geq 23.125$ mm

[d] IOP $\geq 15$ mm Hg

[e] Amount of surgery with above group (7.0 mm: 5.5 mm + above)

[f] Ray-tracing mode.

Three months after surgery, when AL was $\geq$23.125 mm, spherical decreased significantly by -2.013 one week after surgery compared to AL <23.125 mm ($p$ <0.001). Furthermore, the amount of surgery in the above group (7.0 mm: 5.5 mm + above) increased by +0.830 3-months after surgery compared to the 5.0 mm: 4.0 mm + 6.0 mm: 5.0 mm group. In the cylinder, females increased by +0.830 more than males. ($p$ = 0.025). In total corneal astigmatism, flat, steep, and mean K, when AL was $\geq$23.125 mm, all 3 variables also decreased significantly by -0.919, -1.176, and -1.040 a week after surgery compared to AL < 23.125 mm ($p$ = 0.008, $p$ = 0.003, and $p$ = 0.004). Total corneal astigmatism, in patients $\geq$8 years, increased significantly by +0.512 1-week after surgery compared to those <8 years of age ($p$ = 0.010); females decreased -0.446 more than males. ($p$ = 0.011). In HOAs, total RMS ($\mu$m), in patients $\geq$ 8 years, increased significantly by +0.457, 1-week after surgery compared to those <8 years of age. ($p$ = 0.006); that in females decreased by -0.363 as compared to males ($p$ = 0.013) (Table 1).

## Discussion

In this study, we elucidated novel changes of HOAs in children who had undergone R&R surgery for IXT. To date, almost all studies that have evaluated postoperative keratometric and refractive changes merely focused on conventional refractive status; our findings suggest a new concept for post-surgery visual disturbance in strabismus surgery.

Refractive changes after strabismus surgery have been studied by many researchers, but their results are still controversial. A previous study reported significant myopic shifts at 1 month after medial rectus recession [6]. In addition, others reported that R&R surgery continuously affected spherical equivalent (SE: spherical + 1/2 cylinder), with myopic shifts occurring until 3-months postoperatively [9]. The present study was in agreement with these previous results, likely because the average age of the patients of this study was 8.6 years old, which is the age at which myopia progresses physiologically [10, 11]. As for astigmatism after horizontal rectus muscle surgery, with-the-rule (WTR) astigmatism, it has been frequently observed, but most of these studies reported the mean change in astigmatism using keratometry or corneal topography and showed that astigmatic changes recovered during a long-term follow-up period [12, 13]. These studies interpreted most changes in astigmatism to be related to alterations in the corneal curvature. This corneal change is presumed to be due to the tension transmitted to the cornea by pulling the sclera during lateral rectus muscle recession and medial rectus muscle resection [6, 13–15]. Our results agree with this theory, and the simulated and total corneal power evaluations of our analysis showed increasing WTR astigmatism. However, total corneal astigmatism changes, which were measured by ray-tracing analysis, after 1-week and 3-months showed quantitatively different patterns [16]. Considering posterior astigmatism, preoperative total corneal astigmatism was smaller than that evaluated by simulated keratometry. Although the amount of astigmatism was nearly recovered in simulated analysis, total corneal astigmatism was higher than that of simulated astigmatism at postoperative 3-month evaluation. Therefore, we can assume that it takes longer than the period expected by conventional topography to recover from perioperative astigmatic changes during horizontal muscle surgery.

In terms of second-order aberrations, defocus temporarily changed to a negative value (myopic defocus) at 1-week postoperatively, although it returned to near baseline and slightly increased compared to baseline after 3-months. Oblique astigmatism did not change because the vector of the force generated by the R&R surgery acts horizontally. A second-order aberration was a kind of a spherocylindrical error; the change of defocus proved postoperative myopic change came not from age-related myopic progression, but the temporary alteration of corneal curvature by rectus muscle repositioning.

Our study showed that third and fourth-order aberrations with critical effects of corneal state, trefoil, SA, secondary astigmatism, and quadrafoil only changed at 1-week postoperatively, after which they returned to baseline, although the other values showed no changes. In particular, the RMS, which represented the sum of the total HOA values, showed changes at 1-week postoperatively, and a return to nearly baseline 3-months postoperatively, although not to the normal range, thus not affected by corneal and conjunctival swelling immediately following surgery, but corneal and scleral distortion caused by horizontal rectus muscle tightening and release. A change of SA is a well-known factor in decreased contrast sensitivity with significantly increased halo and glare. In addition, coma is associated with vertical deviation of the center, tilt, and double vision in ray. These changes of SA and coma result in deterioration of visual quality that cannot be noticed by clinicians on regular visual acuity test and refraction [17, 18].

In children, if the required precautions are not taken or if refractive changes are not recognized postoperatively, strabismus surgery may result in serious complications, such as changes in diplopia, recurrences of strabismus, and aggravation of amblyopia.

Especially, a few reports suggested the possibility of HOA causing or exacerbating amblyopia [19–21]. Patients who undergo R&R are typically children who find it difficult to verbalize experiences of visual disturbances. Therefore, clinicians need to be aware of the possibility of changes in HOAs before surgery in young individuals, and regular check-up for postoperative HOA would be desirable for amblyopia.

In our multivariable model analysis, long AL was a factor related to higher myopic change, and K to reflect less astigmatism change. Previous studies have reported that the thickness of the posterior sclera is negatively correlated with AL [22, 23]. Accordingly, patients who have longer ALs might have low scleral thickness and increased flexibility. Previous reports showed no correlation between differences in the extent of surgery and refractive power [4, 24]. However, Denis et al. [3] reported that changes in the extent of recession and astigmatism were inversely related after surgery. The present study showed that a greater amount of surgery only affected spherical power, not cylinder and keratometry. Thus, a greater amount of surgery seemed to be related to sclera and muscle healing state.

Furthermore, older age and female sex affected cylinder values, astigmatism, and RMS. Several previous in vivo studies, using various measurement techniques, have found a thicker anterior sclera in males than in females [25–27]. Significant changes in scleral thickness with age, from childhood to early adulthood, have also been demonstrated; these changes were more prominently observed in locations distal to the scleral spur [28]. Therefore, scleral resistance to tension change arisen by rectus muscle reposition in children seems to be lower than in adults.

One of the main limitations of the present study was the short follow-up period conducted with limited patients who had undergone 3 repetitive Scheimpflug examinations during the follow-up period of only 3 months. Long-term prospective studies may provide further insight into the extent of changes in the cornea related to horizontal muscle surgery. Although corneal HOAs are affected by the ocular surface state, we could not obtain tear break-up time (TBUT) and tear meniscus height (TMH) due to the retrospective study design. Future studies should include detailed investigations of changes in TBUT and TMH.

## Conclusions

R&R surgery induces transient myopic deviation and temporarily increases WTR astigmatism as assessed through total corneal astigmatic evaluations. The total RMS and SA also changed 1-week after the operation but returned to baseline by 3-months in HOA analysis. These

changes are significantly related to AL, the amount of surgery, age, and sex, which affected the amount of change in astigmatism due to differences in the scleral state of each patient at each time point. Clinicians need to be aware of the possibility of changes in refraction and HOAs related to strabismus surgery in young individuals who may not be able to verbalize changes in vision, particularly those with amblyopia.

## Supporting information

**S1 Table. Classification of the extent of surgery by exodeviation prism diopter.** LR: lateral rectus; MR: medial rectus; PD: prism diopter.
(DOCX)

**S2 Table. Changes in the IOP, refraction, corneal power, LOAs, and HOAs compared to the preoperative baseline.** IOP: intraocular pressure; LOA: lower-order aberrations; HOA: high-order aberrations.
(DOCX)

## Acknowledgments

We would like to thank Editage (www.editage.co.kr) for English language editing.

## Author Contributions

**Conceptualization:** Dong Cheol Lee, Se Youp Lee, Jong Hwa Jun.

**Data curation:** Dong Cheol Lee, Jong Hwa Jun.

**Formal analysis:** Dong Cheol Lee.

**Funding acquisition:** Dong Cheol Lee.

**Investigation:** Dong Cheol Lee.

**Methodology:** Dong Cheol Lee, Se Youp Lee, Jong Hwa Jun.

**Project administration:** Dong Cheol Lee, Jong Hwa Jun.

**Resources:** Dong Cheol Lee, Se Youp Lee.

**Software:** Dong Cheol Lee, Jong Hwa Jun.

**Supervision:** Dong Cheol Lee, Se Youp Lee, Jong Hwa Jun.

**Validation:** Dong Cheol Lee, Se Youp Lee, Jong Hwa Jun.

**Visualization:** Dong Cheol Lee.

**Writing – original draft:** Dong Cheol Lee.

**Writing – review & editing:** Dong Cheol Lee, Jong Hwa Jun.

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
