## [Decision Letter · Decision Letter 0]

14 Dec 2021

PONE-D-21-33296Alterations of lower- and higher-order aberrations after unilateral horizontal rectus muscle surgery in children with intermittent exotropia: A retrospective cross-sectional studyPLOS ONE

Dear Dr. Jun,

Thank you for submitting your manuscript to PLOS ONE. After careful consideration, we feel that it has merit but does not fully meet PLOS ONE’s publication criteria as it currently stands. Therefore, we invite you to submit a revised version of the manuscript that addresses the points raised during the review process.

ACADEMIC EDITOR:The reviewers have raised serious concerns regarding the study design and whether it was a retrospective or prospective one. The statistical analysis is inappropriate in many parts. In addition, the conclusions are not well supported by the results.

We look forward to receiving your revised manuscript.

Kind regards,

Ahmed Awadein, MD, Ph.D, FRCS

Academic Editor

PLOS ONE

“This research was supported by the Bisa Research Grant of Keimyung University in 2020.”

We note that you have provided information within the Acknowledgements Section. Please note that funding information should not appear in the Acknowledgments section or other areas of your manuscript. We will only publish funding information present in the Funding Statement section of the online submission form.

“This study was supported by the Bisa Research Grant of Keimyung University in 2020 (no URL; no grant number, recipient: DCL). The funder had no role in study design, data collection and analysis, decision to publish, or preparation of the manuscript.”

Reviewers' comments:

Reviewer's Responses to Questions

**Comments to the Author**

1. Is the manuscript technically sound, and do the data support the conclusions?

Reviewer #1: Yes

Reviewer #2: Partly

2. Has the statistical analysis been performed appropriately and rigorously? 

Reviewer #1: Yes

Reviewer #2: No

3. Have the authors made all data underlying the findings in their manuscript fully available?

Reviewer #1: Yes

Reviewer #2: Yes

4. Is the manuscript presented in an intelligible fashion and written in standard English?

Reviewer #1: Yes

Reviewer #2: Yes

5. Review Comments to the Author

Reviewer #1: 1.Altough retrospective, why did you chose to do unilateral R&R for IXT and not bilateral LR recession and was comittance affected?

2. Being a retrospective study, is it standard procedure to follow up astigmatism is your IXT patients using by Scheimpflug Analyzer ? Were both eyes examined and was there a difference ?

3. Were all your patients suffering from unilateral ambylopia?

4. Line 113 please clarify incision at fornix used for what.

5. Line 206 what is meant by " in spherical" ?

6. Line 208, 209 please clarify the surgical groups and what was done for each?

7. line 213 do you < instead of "at 23"

8. Line 279 what is "cross eye angle"?

Reviewer #2: This paper studies the effect of unilateral horizontal rectus muscle R&R surgery in IXT patients on lower and higher order corneal aberrations, with analysis of some demographic and clinical factors that could be related to post strabismus surgery visual disturbances. there are some major and other issues on reviewing the article;

Major issues:

1. Authors have stated that their study is a retrospective one. Nevertheless, they have presented preoperative clinical data that is not routine in preoperative strabismus evaluation; such as axial lens measurement and assessment of ocular aberrations using Scheimpflug analyzer. For me and regarding the data presented in this manner, this study is a prospective one, or do authors have reasonable explanation for performing these unusual clinical and investigative tests before ordinary strabismus surgery?

2. using paired t-test to compare multiple results before and at multiple time points after surgery is not appropriate. ANOVA test is a more reliable option. Statistical analysis of the data presented needs major revision

Other issues:

INTRODUCTION:

3rd paragraph, line 62: Diplopia differs from blurred vision. Post strabismus surgery diplopia has many causes; most commonly overcorrection, muscle slippage, muscle loss. Authors here , in my opinion want to refer to visual disturbance after strabismus surgery, but should be cautious when using the term diplopia.

METHODS:

1. 2nd paragraph, line 83: why authors choose 16 y as age limit in their retrospective analysis? This should be clarified.

2. 2nd paragraph, line 87: were those patients amblyopic before Sx? and if so, what were the causes of Amblyopia? refractive/ anisometropia/starbismic?? and what was the impact of amblyopia on surgical decision? was the amblyopia treated before surgery or not? All these issues need to be clarified?

3. MEASURMENTS; line 97; "deviation angles" this structure is awkward. use angle of exodeviation instead.

4. MEASURMENTS; lines 112,113: Do authors mean that surgery was performed by doing LR recession through fornix incision and MR resection through limbal incision? If so, this part should be rephrased.

RESULTS:

1. 1st paragraph; line 133: "the amount of surgery (5.0 mm: 4.0 mm + 6.0 mm: 5.0 mm versus

above). This is totally confusing. The amount of surgery should be provided in a clear manner to all readers.

2. Changes in measurements postoperatively; lines 138: "All preoperative baseline parameters and the changes over time for IOP, refraction, simulated or total corneal power, LOAs, and HOAs are summarized in the S1 Table." when reviewing S1 table, no baseline data has been included.

3. In general, results are expressed in an extremely confusing manner. authors should do more effort to clarify and simplify the results obtained.

DISCUSSION:

1. line 277; "In children, if the required precautions are not taken,.." what are these precautions?

2. line 279; " complications, such as changes in the cross-eye angle". what authors mean by changes in cross eye angle?

CONCLUSION:

line 317: "Clinicians need to be aware of the possibility of changes in refraction and HOAs related to strabismus surgery in young individuals who may not be able to verbalize changes in vision, particularly those with amblyopia".

How authors reach this conclusion? From data presented, I don't see facts that support this conclusion.

6. PLOS authors have the option to publish the peer review history of their article (what does this mean?). If published, this will include your full peer review and any attached files.

Reviewer #1: No

Reviewer #2: No

---

## [Author Response · Author response to Decision Letter 0]

24 Jan 2022

Review Comments to the Author

Reviewer #1: 

1.Altough retrospective, why did you chose to do unilateral R&R for IXT and not bilateral LR recession and was comittance affected?

Thanks for the valuable comment. First, our clinic usually conducts unilateral R&R on patients with IXT. In addition, under the assumption that the change in astigmatism can be affected differently by recession or resection the muscle during surgery, we tried to reduce the bias by limiting to unilateral R&R surgery, which is often performed at our hospital, and looking at the changes at each time point after surgery. Thank you once again for this pertinent comment.

2. Being a retrospective study, is it standard procedure to follow up astigmatism is your IXT patients using by Scheimpflug Analyzer ? Were both eyes examined and was there a difference ?

Thank you for the valuable comment. First, refraction, retinal examination, and topography are performed among patients with strabismus as preoperative examinations for children who think they need surgery. These are mainly young children who are at a risk of amblyopia, and retinal examination with axial length is also tested as part of this examination. Topography is inspected to check the changes in corneal shape before and after R&R surgery, which is explained to parents. The topography imaging machine is ‘gallilei G4’, and it has a Scheimpflug analyzer function for topography inspection owing to its characteristics; hence, the results are displayed together. Even now, pre-surgical examination and post-operative topography are performed in children with strabismus. We have summarized the above-mentioned information and mentioned it in the methods section. In addition, since only one eye was operated, the results were compared only for one eye. Thanks once again for the pertinent comment.

Among these patients, we only enrolled those patients who underwent Scheimpflug photography at each visit to evaluate astigmatism and accurately analyze the refractive status of amblyopia --� Among these patients, most of them were assessed for refractive error with topography and retinal examination with axial length (AL) before and after R&R surgery to detect the degree of the problem. In this study, we only enrolled patients who underwent Scheimpflug photography with topography at each visit to evaluate astigmatism and accurately analyze the refractive status.

3. Were all your patients suffering from unilateral ambylopia?

Thank you for the valuable comment. Of the 85 children who underwent surgery, 12 had amblyopia, and the remaining 73 had normal vision. As is well known, many children with strabismus complain of visual disturbance after surgery; moreover, children with strabismus and refractive amblyopia may be particularly vulnerable. As per your suggestion, We have omitted the word 'amblyopia' from the paragraph since it can result in confusion about amblyopia in line 85.

Among these patients, we only enrolled patients who took Scheimpflug photography at each visit to evaluate astigmatism and accurately analyze the refractive status of amblyopia --� Among these patients, most of them were assessed for refractive error with topography and retinal examination with axial length (AL) before and after R&R surgery to detect the degree of the problem. In this study, we only enrolled patients who underwent Scheimpflug photography with topography at each visit to evaluate astigmatism and accurately analyze the refractive status.

4. Line 113 please clarify incision at fornix used for what.

Thank you for the valuable comment. I have revised the the following sentence in Line 114-5 in the manuscript. Thank you.

‘All surgeries were conducted by a single surgeon (LSY) who performed a limbal incision at the medial rectus muscle resection and an incision in the fornix.

All surgeries were conducted by a single surgeon (LSY) who performed a limbal incision at the medial rectus muscle resection and an incision in the fornix at the lateral rectus muscle recession.

5. Line 206 what is meant by " in spherical" ?

Thank you for the valuable comment. The word ‘in spherical’ is a typographical error, which should be deleted from the text. Thanks once again for the pertinent comment.

6. Line 208, 209 please clarify the surgical groups and what was done for each?

Thank you for the valuable comment. As you have suggested, the amount of surgery according to the IXT angle before surgery has been added to the table below. (Table S1). Furthermore, we have added the following sentence in the Methods section and add Table S1 to clarify its meaning. Thanks once again for the pertinent comment.

Furthermore, patients were divided into two groups based on the extent of the surgery. Patients in the first group underwent unilateral 5.0-mm lateral rectus resection (LR res.), followed by 4.0-mm medial rectus resection (MR res.) and 6.0-mm LR res. with 5.0-mm MR res., whereas patients in the second group underwent 7.0-mm LR res. with 5.5-mm MR res. with a greater extent of the sugery (S1 Table ).

S1 Table. Classification of the extent of surgery by exodeviation prism diopter

Exodeviation angle Extent of surgery (LR recession/MR resection) Number of patients

20 PD 5.0 mm/4.0 mm 14

25 PD 6.0 mm/5.0 mm 37

30 PD 7.0 mm/5.5 mm 17

35 PD 7.5 mm/6.0 mm 10

40 PD 8.0 mm/6.5 mm 6

50 PD 10.0 mm/7.0 mm 1

Total 85

LR: lateral rectus; MR: medial rectus; PD: prism diopter

7. line 213 do you < instead of "at 23"

Thank you for the valuable comment. It is more precise to replace ‘at’ with ‘<’ in the text. I have edited the text accordingly. Thanks once again for the pertinent comment.

8. Line 279 what is "cross eye angle"?

Thank you for the valuable comment. Revising ‘changes in the cross-eye angle’ to ‘diplopia’ seems easier to understand. I have corrected the text accordingly. Thanks once again for the pertinent comment.

Reviewer #2: 

This paper studies the effect of unilateral horizontal rectus muscle R&R surgery in IXT patients on lower and higher order corneal aberrations, with analysis of some demographic and clinical factors that could be related to post strabismus surgery visual disturbances. there are some major and other issues on reviewing the article;

Major issues:

1. Authors have stated that their study is a retrospective one. Nevertheless, they have presented preoperative clinical data that is not routine in preoperative strabismus evaluation; such as axial lens measurement and assessment of ocular aberrations using Scheimpflug analyzer. For me and regarding the data presented in this manner, this study is a prospective one, or do authors have reasonable explanation for performing these unusual clinical and investigative tests before ordinary strabismus surgery?

Thank you for the valuable comment. First, refraction, retinal examination, and topography are performed among strabismus patients as preoperative examinations for children who think they need surgery. These are mainly young children who are at a risk of amblyopia, and retinal examination with axial length is also tested as part of this examination. Topography is inspected to check the changes in corneal shape before and after R&R surgery, which is explained to parents. The topography imaging machine is ‘gallilei G4’, and it has a Scheimpflug analyzer function for topography inspection owing to its characteristics; hence, the results are displayed together. Even now, pre-surgical examination and post-operative topography are performed in children with strabismus. I have summarized the above-mentioned information and mentioned it in the methods section. Thanks once again for the pertinent comment.

Among these patients, we only enrolled patients who underwent Scheimpflug photography at each visit to evaluate astigmatism and accurately analyze the refractive status of amblyopia --� Among these patients, most of them were assessed for refractive error with topography and retinal examination with axial length (AL) before and after R&R surgery to detect the degree of the problem. In this study, we only enrolled patients who underwent Scheimpflug photography with topography at each visit to evaluate astigmatism and accurately analyze the refractive status.

2. using paired t-test to compare multiple results before and at multiple time points after surgery is not appropriate. ANOVA test is a more reliable option. Statistical analysis of the data presented needs major revision

Thank you for the valuable comment. You are correct; however, if you look at the results of our thesis, we compared the results of the changes before and 1 week after the surgery and the results of the changes before and 3 months after the surgery using a paired t-test, respectively.

In the methods section,

 ‘Paired sample t-tests were performed to evaluate differences in outcomes before surgery and at 1-week and 3-months post-surgery’ � 'Paired sample t-tests were performed to evaluate the differences in the outcomes before surgery and at 1 week post-surgery, and before surgery and 3 months post-surgery'. 

Thank you.

Other issues:

INTRODUCTION:

3rd paragraph, line 62: Diplopia differs from blurred vision. Post strabismus surgery diplopia has many causes; most commonly overcorrection, muscle slippage, muscle loss. Authors here , in my opinion want to refer to visual disturbance after strabismus surgery, but should be cautious when using the term diplopia.

Thank you for the valuable comment. As you have suggested, visual disturbance after strabismus surgery is better. On Line 62, ‘diplopia or blurred vision’ has been replaced with “visual disturbance.” Thanks once again for your valuable comments.

METHODS:

1. 2nd paragraph, line 83: why authors choose 16 y as age limit in their retrospective analysis? This should be clarified.

Thank you for the valuable comment. First, this paper is a retrospective study, and most of the patients who underwent topography before surgery were children before they reached adulthood, and among them, the oldest patient was 16 years old. As per your suggestion, we have revised the sentence as mean ± SD. Thanks once again for the pertinent comment.

In line 83, We retrospectively reviewed the medical records of all patients aged 16 years and under who had undergone R&R surgery for IXT repair � We retrospectively reviewed the medical records of all patients (mean age, 8.64±2.88 years) who had undergone R&R surgery for IXT repair.

2. 2nd paragraph, line 87: were those patients amblyopic before Sx? and if so, what were the causes of Amblyopia? refractive/ anisometropia/starbismic?? and what was the impact of amblyopia on surgical decision? was the amblyopia treated before surgery or not? All these issues need to be clarified?

Thank you for the valuable comment. Of the 85 children who underwent surgery, 12 had amblyopia, and the remaining 73 had normal vision. As is well known, many children with strabismus complain of visual disturbance after surgery, and children with strabismus and refractive amblyopia may be particularly vulnerable. As per your suggestion, we have omitted the word 'amblyopia' from the following paragraph to avoid confusion.

Among these patients, we only enrolled patients who underwent Scheimpflug photography at each visit to evaluate astigmatism and accurately analyze the refractive status of amblyopia --� Among these patients, most of them were assessed for refractive error with topography and retinal examination with axial length (AL) before and after R&R surgery to detect the degree of the problem. In this study, we only enrolled patients who underwent Scheimpflug photography with topography at each visit to evaluate astigmatism and accurately analyze the refractive status.

3. MEASURMENTS; line 97; "deviation angles" this structure is awkward. use angle of exodeviation instead.

Thank you for the valuable comment. As per your suggestion, I have made the necessary corrections to the text. Thanks once again for the pertinent comment.

Deviation angles were measured using the alternate prism cover test for distance (6 m) and near (33 cm) using accommodative targets and the patients’ best optical correction. � The angle of exodeviation was measured using the alternate prism cover test for distant (6 m) and near (33 cm) objects using accommodative targets and the patients’ best optical correction.

4. MEASURMENTS; lines 112,113: Do authors mean that surgery was performed by doing LR recession through fornix incision and MR resection through limbal incision? If so, this part should be rephrased.

Thank you for the valuable comment. I have revised the following sentence and added it to Line 114-5. Thank you.

All surgeries were conducted by a single surgeon (LSY) who performed a limbal incision at the medial rectus muscle resection and an incision in the fornix.

All surgeries were conducted by a single surgeon (LSY) who performed a limbal incision at the medial rectus muscle resection and an incision in the fornix at the lateral rectus muscle recession.

RESULTS:

1. 1st paragraph; line 133: "the amount of surgery (5.0 mm: 4.0 mm + 6.0 mm: 5.0 mm versus

above). This is totally confusing. The amount of surgery should be provided in a clear manner to all readers.

Thank you for the valuable comment. As per your suggestion, the amount of surgery according to the IXT angle before surgery has been added to the table below (Table S1). Furthermore, we have added the following sentence in the Methods section and added Table S1 to clarify its meaning. Thanks once again for the pertinent comment.

We have added the following to lines 115-116: ‘Furthermore, the surgery groups were divided into two groups (recession: resection; 5.0 mm: 4.0 mm + 6.0 mm: 5.0 mm and 7.0 mm: 5.0 mm + above).’

Table S1. Classification of the extent of surgery by exodeviation prism diopter

Exodeviation angle Extent of surgery (LR recession/MR resection) Number of patients

20 PD 5.0 mm/4.0 mm 14

25 PD 6.0 mm/5.0 mm 37

30 PD 7.0 mm/5.5 mm 17

35 PD 7.5 mm/6.0 mm 10

40 PD 8.0 mm/6.5 mm 6

50 PD 10.0 mm/7.0 mm 1

Total 85

LR: lateral rectus; MR: medial rectus; PD: prism diopter

2. Changes in measurements postoperatively; lines 138: "All preoperative baseline parameters and the changes over time for IOP, refraction, simulated or total corneal power, LOAs, and HOAs are summarized in the S1 Table." when reviewing S1 table, no baseline data has been included.

Thank you for the valuable comment. Table S2 indicates the changes compared to before surgery at 1 week after surgery and 3 months after surgery compared to before surgery. We wished to present the change following surgery at a glance by showing the amount of each change and showing statistical significance. Therefore, we have changed the legend of Table S2 as follows.

Table S2 legend has been revised as

Preoperative baseline parameters and changes in IOP, refraction, corneal power, LOAs, and HOAs.-� Changes in the IOP, refraction, corneal power, LOAs, and HOAs compared to the preoperative baseline.

3. In general, results are expressed in an extremely confusing manner. authors should do more effort to clarify and simplify the results obtained.

Thanks for the valuable comment. A large amount of change were observed before and after surgery, including not only high-order aberrations but also topography values. Thus, when writing the results, I have divided them into “Changes in the measurements postoperatively” and “Multivariable analysis,” and tried to express them in the respective figures and tables. Moreover, I tried to clarify the parts that may cause be confusing by correcting the text and adding Table S1. Thanks once again for the pertinent comment.

DISCUSSION:

1. line 277; "In children, if the required precautions are not taken,.." what are these precautions?

Thank you for the valuable comment. First, children's eyesight develops until the age of 6, and if a child with strabismus proceeds with surgery without identifying whether diplopia or refractive changes may occur after surgery, the eyesight or amblyopia cannot be corrected after surgery. It may affect stereoscopic vision; particularly, since children may not be able to express all of their visual disturbance in words, doctors and parents should be aware of the refractive changes before and after surgery.

2. line 279; " complications, such as changes in the cross-eye angle". what authors mean by changes in cross eye angle?

Thank you for the valuable comment. Revising ‘changes in the cross-eye angle’ to ‘diplopia’ seems easier to understand. I have corrected the text accordingly. Thanks once again for the pertinent comment.

CONCLUSION:

line 317: "Clinicians need to be aware of the possibility of changes in refraction and HOAs related to strabismus surgery in young individuals who may not be able to verbalize changes in vision, particularly those with amblyopia".

How authors reach this conclusion? From data presented, I don't see facts that support this conclusion.

Thank you for the valuable comment. As you are aware, HOAs can affect vision with higher order aberrations, which cannot be corrected with glasses. In this study, it was found that normal HOAs had an effect by increasing or decreasing 1 week after surgery, and nearly returned to the pre-operative level after 3 months. Therefore, at 3 months before strabismus surgery, visual acuity that cannot be corrected with glasses could be present due to the changes in HOAs, and children may not be able to express this well in words. Accordingly, the doctor who performs strabismus surgery should be aware of this fact in advance and prescribe eyeglasses to children who need refractive correction from 3 months after the surgery and thereby prevent amblyopia.

---

## [Editor Report · Decision Letter 1]

2 Feb 2022

Alterations of lower- and higher-order aberrations after unilateral horizontal rectus muscle surgery in children with intermittent exotropia: A retrospective cross-sectional study

PONE-D-21-33296R1

Dear Dr. Jun,

We’re pleased to inform you that your manuscript has been judged scientifically suitable for publication and will be formally accepted for publication once it meets all outstanding technical requirements.

Kind regards,

Ahmed Awadein, MD, Ph.D, FRCS

Academic Editor

PLOS ONE
---

## [Editor Report · Acceptance letter]

8 Feb 2022

PONE-D-21-33296R1 

Alterations of lower- and higher-order aberrations after unilateral horizontal rectus muscle surgery in children with intermittent exotropia: A retrospective cross-sectional study 

Dear Dr. Jun:

I'm pleased to inform you that your manuscript has been deemed suitable for publication in PLOS ONE. Congratulations! Your manuscript is now with our production department. 

Kind regards, 

on behalf of

Dr. Ahmed Awadein 

Academic Editor

PLOS ONE